# Evaluating the Visual Metaphors of Financial Concepts through Content Analysis

Awais Malik

Fakultät Wirtschaftswissenschaften, Technische Universität Dresden, 01062 Dresden, Germany; awais.malik@tu-dresden.de

**Abstract:** Adding pictures to instructional materials that are relevant and representational supports meaningful learning. However, it is not always straightforward to generate such pictures, for example, for abstract concepts. It is much easier to make representational pictures of concrete concepts, "table" or "chair", compared to abstract concepts, "loyalty" or "democracy". The field of finance is full of abstract or complex financial concepts, such as pension, market value, and asset valuation—to name a few. How do we then make pictures of such financial concepts that can represent them? In this regard, visual metaphors could provide hints as to how complex financial concepts can be presented in the form of pictures. For this purpose, this study analyzed the representation of complex financial concepts in terms of visual metaphors. Visual metaphors of five financial concepts were selected from the financial learning content online. These included: (1) risk diversification, (2) inflation, (3) compound interest, (4) time value of money, and (5) financial risk. Using the content analysis approach, each of the visual metaphors were analyzed to determine how different features of the given financial concept were mapped onto the visual metaphor, making them representational. Results indicate that visual metaphors could be an effective and creative way to present complex financial concepts in the form of representational pictures.

**Keywords:** financial literacy; visual metaphors; financial education; content analysis

## 1. Introduction

Financial literacy is essential for making informed financial decisions and having a better life. It helps people to effectively manage their finances related to everyday living, education, post-retirement, and other major aspects of life, along with managing external uncertainties (Mandell and Klein 2009). In times where pandemics, wars, and financial crises are recurrent, an individual's inability to make sound financial decisions only exacerbates the situation. The benefits of financial literacy are in many shapes and forms, and also include higher financial inclusion that directly supports the overall economy (Serido 2020; Chhatwani and Mishra 2021). However, given the importance of financial literacy, people fail to have even a rudimentary knowledge of finance (Goyal and Kumar 2021; Lyons and Kass-Hanna 2021). Moreover, financial products such as household mortgages, insurance, and investment options have become highly complex and the inability of people's comprehension is apparent, which leads to dire consequences (Hermansson et al. 2022; Li 2020; Remund 2010; Lusardi 2015).

To address this issue, the information about financial concepts should be presented in an easily understandable form (Neuberger et al. 2022; Paramonovs and Ijevleva 2015; Remund 2010; Eberhardt et al. 2021). One of the ways to do this is to add relevant pictures to financial learning materials. Pictures open up visual cognitive resources of the mind that aid in understanding the concepts better (Winokur et al. 2019). Learning with pictures helps in improving memory (Zormpa et al. 2019), the transfer of skills (Mason et al. 2016), engagement (Li and Xie 2020), and decision making (Kaplan et al. 2016). However, the challenge in finance is that it is complicated to generate relevant pictures of most financial

concepts. This is because finance is filled with complex, abstract concepts, for example asset valuation, inflation, liquidity, and simple and compound interest—to name a few. Not only are such concepts difficult to learn, they are even harder to picture (McRae et al. 2018). Unlike concrete concepts that can be experienced through the senses of sight, smell, sound, taste, or touch that aid in building pictures, abstract concepts are devoid of such perceptual referents. This poses a challenge to generate pictures that are representational o such concepts. Generating a representational picture of something that has a palpable, concrete form is much easier, compared to something that is not directly perceived through our senses.

It is important that pictures are representational or relevant in learning; because, as per the cognitive theory of multimedia learning (CTML) (Mayer 2002), if pictures are relevant to the instructional material, it enhances the generative processing, which is higher engagement in learning (Mayer 2014). However, in cases where the pictures are irrelevant, it may increase extraneous processing, which reduces the effectiveness of learning materials (Parong and Mayer 2021). Abstract concepts, owing to their imperceptible nature, are at a much higher risk of generating nonrepresentational or irrelevant pictures. In relation to this, visual metaphors could be used as relevant pictures in finance, because they provide flexibility and room for creativity to visually present the mental models (Schwartz 2020). Moreover, metaphors are known as a particularly potent method for comprehending complex, abstract concepts (Jensen 2006; Jamrozik et al. 2016). They have been extensively studied in the education field, which consistently shows that they help in learning difficult concepts (Rieber and Noah 2008; Franzoni et al. 2020; Alyahya 2018). However, the financial literacy field is underexplored in terms of studying visual metaphors for improving financial learning. Although metaphors are informally used in financial learning, this kind of usage is similar to using metaphors in everyday life situations for explaining difficult topics. The formal recognition of visual metaphors in the field of finance, as a method of representing complex concepts, and further study in refining the metaphors for the increased effectiveness of financial education is very scarce.

With this backdrop, this study is a starting point that conducted content analysis of existing visual metaphors related to financial concepts that are present online on different educational websites produced by educators or professional artists. The aim is to show how complex financial concepts can be represented through visual metaphors. There are two reasons for choosing an online context. One, informal learning on the internet is rising in the field of finance (Rudeloff 2019), where a considerable number of people, especially younger adults, regularly use the internet to self-learn about financial topics. Two, digitization provides greater opportunities to present information in many different ways, so there are many webpages teaching financial concepts by applying various creative ideas, including visual metaphors. This study could provide direction and hints for further research in developing and/ or using visual metaphors for learning complex, abstract financial concepts.

*Conceptual Metaphor Theory and Visual Metaphors*

It was traditionally believed that metaphors were confined only to literary purposes. However, a unique explanation of metaphor was given in the transforming book, *Metaphors We Live By* written by (Lakoff and Johnson 1980) that changed the whole notion of metaphors and introduced the conceptual metaphor theory (CMT). They introduced a novel perspective, in which metaphors are viewed as a cognitive process where one domain is conceptualized in terms of another, called mappings. It works by applying the features of one domain (source) to better understand an abstract, complex domain (target) based on the similarities between the two (Holyoak and Stamenković 2018). For example, in the "love is a journey" metaphor (Lakoff 1993), the source domain is "journey" that is used to understand the target domain "love". Lakoff (1993, p. 207) analyzed the mappings from source to target domain as: "the lovers correspond to travellers", "the love relationship corresponds to the vehicle", "the lovers' common goals correspond to their

common destinations on the journey", and "difficulties in the relationship correspond to impediments to travel".

A visual metaphor is a particular type of metaphor whose medium is pictorial. As with any metaphor, the features of the visual metaphor's source domain are mapped onto the features of the target domain. The only difference is that in visual metaphors, the target and/or source domain can be represented using visuals (Ortiz 2011). Visual metaphors are known to be an effective approach to depicting and explaining complex concepts (Forceville 2008; Cornelissen et al. 2011). As mentioned, metaphors are driven from the concrete domains that provide higher perceived relevance compared to abstract visualization. In doing so, it also supports the transfer of knowledge (Eppler 2003). Visual metaphors engage the learners in the content because such depictions are not straightforward. That lets the individuals to think deeper and find about the possible meanings (McQuarrie and Mick 1999). Visual metaphors provide an effective way to emphasize specific attributes of the concepts and thus aid in framing the issues.

## 2. Methodology

### 2.1. Research Design

The aim of the study is to show how abstract financial concepts can be represented in the form of visual metaphors. Thus, the study applied the qualitative content analysis approach to analyze the mappings between visual metaphors (source domain) and financial concepts (target domain), which were available online on educational webpages. For this purpose, the selected financial concepts were those that are usually needed for planning or making investments. These included five concepts: (1) risk diversification, (2) inflation, (3) compound interest, (4) the time value of money, and (5) financial risk. The reasons for selecting investment-related concepts are that it is one of the major activities of the finance field, and it includes future planning along with managing resources that are at hand today, for which it requires high discipline and responsible behavior (Carlin and Robinson 2012). For these reasons, it is crucial for people to understand such concepts related to investments.

### 2.2. Sample and Procedure

The sample was selected following the purposeful sampling technique. The visual metaphors of the selected financial concepts were retrieved from the Google search engine. In total, five visual metaphors were selected, one for each concept. The reason for selecting one for each and not more was to avoid redundancy. Each concept was searched separately with different keywords related to that particular concept. For instance, in the case of the concept of inflation, keywords such as "understanding inflation", "learning about inflation", "what is inflation?", or other related words were used to search for its visual metaphor. This same process was repeated for all of the five concepts.

The Google search engine presents various options for search, typically "Images", "News", "Books", and so on. One of the options is "All"; this was selected because it provides clear titles and a brief description of the webpages, which helped in assessing whether the webpage was educational or not. On "All", the search results present the links to other webpages. On each search query, Google presents several pages of results, only the results of the first page were considered. The webpages were visited to search for visual metaphors. It was required that the content should only be educational where the concept is explained comprehensively. For instance, if it was an advertisement or news, then those results were eliminated. This was cued through the information on the webpages, and their focus on learning instead of, for instance, selling a product. If a webpage was other than educational, but it included a visual metaphor, it was not selected. However, if a webpage was learning-based as well as including a visual metaphor, then that was taken into account. Following these criteria, as soon as a visual metaphor was found, a further search was stopped for that financial concept and moved on to searching for the next one.

### 2.3. Analysis Approach

The analysis adhered to the guiding principle of metaphorical formula A IS B (Bounegru and Forceville 2011). In this, one conceptual domain is comprehended in terms of another conceptual domain. B is a source domain that contrives and forms the target domain A. One of the examples that Lakoff and Johnson (1980, pp. 4–5) discussed in their book is: ARGUMENT IS WAR to illustrate how an idea is understood using different conceptual domains. They showed that the source domain in this metaphor is WAR to construe the target domain ARGUMENT. The correspondences or mappings of the conceptual metaphor, ARGUMENT IS WAR, refers to two opposing parties battling with each other to defend their own positions and attack the other. The writers view its reflections in expressions that included, among others, "Your claims are indefensible", "He attacked every weak point in my argument", and "I demolished his argument". Against this background, Forceville (2002) proposed the following structure to identify and interpret metaphors:

1.　Which are the two terms of the metaphor and how do we know?
2.　Which one is the metaphor's target domain, which one is the metaphor's source domain, and how do we know?
3.　Which features can/should be mapped from the source domain to the target domain, and how is their selection decided upon?

In this paper, these questions are followed to inform the analysis of financial concepts' visual metaphors. These questions have been used by many prior studies to interpret and analyze different types of metaphors, including visual metaphors. Crawford and Juricevic consistently used this structure to interpret visual metaphors in their several studies (Crawford and Juricevic 2016, 2018, 2020, 2022). Coëgnarts and Kravanja (2012) in developing a theoretical framework for analyzing the structural–conceptual metaphors and image metaphors, proposed these three questions and developed three additional, specific to the film context.

The answers to the first two questions apply to all the selected visual metaphors in this study, but the answers vary for each one in the third question. Regarding the first question, one term of the metaphor is the financial concepts, because these are the elements of analysis, and another term is the components of the visuals. The second question then moves forward to determining the directionality after establishing the two terms. In this case, the financial concepts are the metaphor's target domain, and components of the visual metaphors are the source domain. The target domain was recognized on the basis of the webpages' content that was related to a given financial concept. In addition, the source domain became known because visuals were used by the author(s) of webpages to explain the financial concepts. This was one of the reasons for selecting educational or learning-focused webpages where the incorporation of visuals is usually for the purpose of explaining the learning content.

Further, the third question is about defining the mappings of the source domain onto the target domain and its selection criteria. Given that the context was finance, only those features that were related to the financial concepts could be mapped from the source domain to the target domain. The definition of these financial concepts was based on the following sources: (1) Gitman et al. (2015), and (2) Zutter and Smart (2019), which were used as the criteria to judge features in these concepts. Within these bounds, the author qualitatively interpreted the correspondences between financial concepts and visual metaphors by applying the metaphorical formula A IS B that is discussed above. Regarding the structure, in the analysis part of visual metaphors, financial concepts were first defined in each of the segments followed by the detailed analysis of the mappings. The issues raised in this third question are discussed in detail in the next section.

Finally, it is important to note, the analysis of visual metaphors discussed in this paper might have multiple interpretations, which is common in studies of interpreting metaphors. However, the scope of this study is not to present a so-called "correct" view of analysis; the purpose is to show, by explaining the linkages of two different domains, how complex financial concepts can be represented through visual metaphors. In fact, it is a unique

aspect of metaphors to have multiple interpretations (Lazard et al. 2016). This is led by their indirect way of unfolding information (Thibodeau et al. 2019). As a result, this incongruity has the potential of gathering learners' attention and stimulating their interest (Tay 2020). Additionally, given that the selected visual metaphors are not purpose-built, they may or may not capture all essential aspects of the financial concepts. Thus, the interpretation of source and target domains is confined to the extent that the visual metaphor of a given financial concept has elements related to the concept.

## 3. Analysis of Visual Metaphors of Financial Concepts

### 3.1. Risk Diversification

Risk diversification is related to the broadening of investments in different types of industries and investing in alternatives in a portfolio. It is also known as a "risk management strategy" of a collection of different types of investments. The rationale for such variety is to reduce the exposure to risk, as having limited financial resources in multiple investment options poses less risk compared to putting all money into one investing option. This is because if one invests all the money in one investment alternative, such as investing in the stocks of one particular company, and if those stocks go down, then all of the money might be lost. Alternatively, if investing the same amount of money in the stocks of several companies in distinct industries, then it is less likely that if one industry performs poorly, the others will also perform the same way. This is one way of managing it, but there is no fixed rule; the main idea is to diversify the investments in a portfolio.

Figure 1 presents this concept in a visual metaphor on the basis of a verbal metaphor of "don't put all your eggs in one basket". This verbal metaphor generally means that one should not rely on putting all efforts and/or resources in one area to avoid the risk of losing everything. This has been visualized in the financial context where the visual metaphor shows the setting of a poultry farm and two situations. One is when the farmer drops the basket containing all the eggs, and in the second, the farmer is putting eggs in different baskets. There are several features that have been covered in this visual metaphor of the "risk diversification" concept. The money is presented in terms of eggs, and the basket in terms of investment options in which the money is invested. In Figure 1a, it shows that the man bumped into a rock placed on the floor and consequently drops his basket of eggs, and at the back the chickens are shouting and crying. All the eggs in one basket shows all the money in one investment option. The bump to the rock refers to the risk of losing money on that investment option. The shouting and crying of the chickens at the back suggest that this is money that the farmer earned with the help of other people, his family, business partners, or employees, who are not happy about losing all the money in just one stumble. In addition, the face of the farmer himself is shown sad and uttering the words of sorrow "oh no my eggs!" representing the grief, which is the emotional aspect of it (Ito et al. 2019).

In Figure 1b, it shows the solution to the risk of losing all money. Here the farmer puts eggs in different baskets representing the diversification of investments. Putting eggs (money) in different baskets (investment options) will reduce the risk of losing all eggs (money) even if one basket falls. However, since one of the major ways to diversify a portfolio is by putting assets in areas that are not correlated to one another, showing the baskets in different colors or shapes would have represented the diversification of investments much better. Further, the varied number of eggs in each basket may represent two things; one, the money is invested unequally in each investment option, and second, the performance of each option (basket) differs, wherein the higher performance means more money (more eggs), and the lower performance means less money (less amount of eggs). The chickens' beaks are closed and tears not rolling down, and the farmer's face is also smiling, showing a content state of mind. This represents the concept that if investments are diversified, then the risk of losing money is fairly low, and lower risk leads to a healthier state of mind.

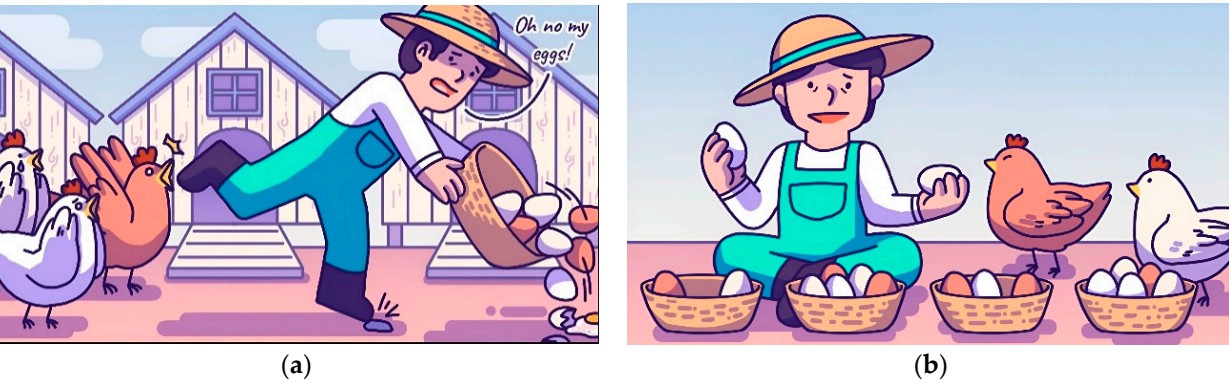

**Figure 1.** (**a**) (left) and (**b**) (right): Visual metaphors of the "risk diversification" concept. Source: www.thesimplesum.com. Available online: http://bit.ly/3Heb72U (archived on 23 January 2023).

### 3.2. Inflation

Inflation is known as an increase in the prices of goods and services. The escalation of prices refers to the devalue of each unit of currency that can be used in exchange for fewer goods or services compared to prior periods. Such a phenomenon is known as the purchasing power of money that declines over time due to the rise in prices. One of the major causes of inflation is an increase in money supply in the economy. The money supply is usually increased by printing more money, an increase in public spending, deprivation of the exchange rate, and other reasons. The opposite of inflation is called "deflation", which is the reduction in prices that causes an increase in purchasing power. In economics, people's level of living is directly related to their purchasing power, which outlines the affordability of a variety of goods and services that people can afford to buy. This relationship between inflation and purchasing power is inversely proportional; if one increases, the other decreases. Inflation is represented by a single value that is often called the inflation rate. This single value accounts for the average price level of goods and services in an economy over a given period of time. Inflation usually has a negative impact on investments, especially ones with long-term holdings.

Figure 2 shows the visual metaphor of the target domain that is the concept of inflation. In this illustration, several specific components of the concept of inflation are covered. It depicts a big dollar sign in the form of a balloon that is attached to an air pump blowing air into it. The dollar sign represents money or other financial resources that can be used to exchange goods and services. The air inside this balloon indicates the value of money. The increasing size of the dollar sign balloon due to the air postulates the increasing rate of inflation. The weightless nature of air is mapped onto the declining value of money. The increasing size of the balloon represents that, since inflation is increasing, it will cause people to need a large amount of money to buy the same goods and services. It can be interpreted as: although the absolute amount of money needed is greater, the value of it is weightless like air. The two lines on each edge of the dollar sign show that the balloon is continuously enlarging in size, which portrays the uninterruptedly increasing rate of inflation. It demonstrates that even if there is a huge stack of money, its value is decreasing because of inflation. A balloon shines when it is fully blown. Similarly, here the dollar sign balloon has a shine as well, specifying that inflation has reached its peak. On the air pump there is a meter that shows a particular numeric value that aids in knowing what amount of air has been filled into the balloon. This meter symbolizes the single value through which inflation is measured; the inflation rate. If the needle on the meter goes towards the higher side, the air in the balloon will also increase. In the same way, a higher rate means higher inflation and vice versa.

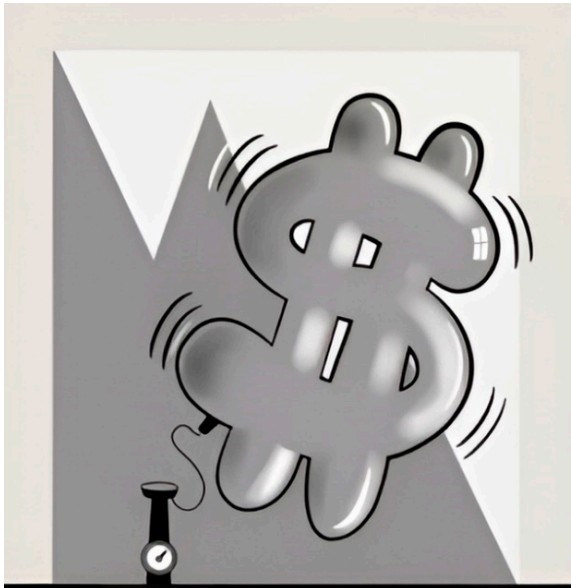

**Figure 2.** Visual metaphor of the "inflation" concept. Source: www.investopedia.com. Available online: http://bit.ly/3HesRLA (archived on 23 January 2023).

### 3.3. Compound Interest

Compound interest is the addition of interest to a loan or deposit. It is calculated on the principal amount and also on the interest that has accumulated from prior periods. It is a type of interest that works as reinvesting interest. Since compound interest calculates interest accumulated in preceding periods, it grows at an exceedingly high rate. This is in contrast to simple interest that is only calculated on the principal amount and does not consider the accumulated interest from previous periods. For example, a deposit of USD 100,000 that is calculated on a simple annual interest of 5% for ten years will earn USD 50,000 at the end of the whole term. Meanwhile, investing USD 10,000 compared to USD 100,000 at a 5% annual compound interest for the same period of time, will reap about USD 63,000. The rate of compound interest also depends on the number of periods; the greater the number of periods, the greater will be the compound interest. If the compounding period is monthly instead of annually, then the amount would be even greater. The negative aspect of compound interest is that it also works against people when taking loans at high interest rates, which is usually the case with credit card debt. In such cases, the burden of paying back the loan becomes gigantic, because the loaned amount is increasing at an exorbitantly high rate.

Figure 3 displays the metaphor of the snowball that becomes bigger in size when it rolls down a snowy mountain or a downwardly inclined surface full of snow. In the figure, there are five different scenes. In the first scene, it shows the person making a snowball and getting ready to throw it down. This scene symbolizes generating an initial deposit or principal amount yourself that will be deposited to earn compound interest. The tongue sticking out represents engagement, dedication, and excitement, as body parts are recognized for communicating in metaphors (Morrow 2013). This demonstrates that the initial deposit should be generated on your own. It may be through the skill that you learned or the knowledge you gained, so you have to use those resources with hard work and devotion. In the second scene, the person finally throws the snowball downwards, which portrays the depositing of money for earning compound interest. Compound interest is depicted in this visual metaphor by the way the ball is thrown downwards on a snowy surface. The rolling down of a snowball is automated in the sense that it happens without the interference of the one who threw the snowball. Moreover, the ball travels down at a fast speed and becomes bigger quickly. This phenomenon is mapped onto the compound interest, which gets added to the initial deposit without the interference of the one who

deposits the money, because the process is automated. The other way of saying it would be that the money grows itself and at a faster rate when it is put on a compound interest.

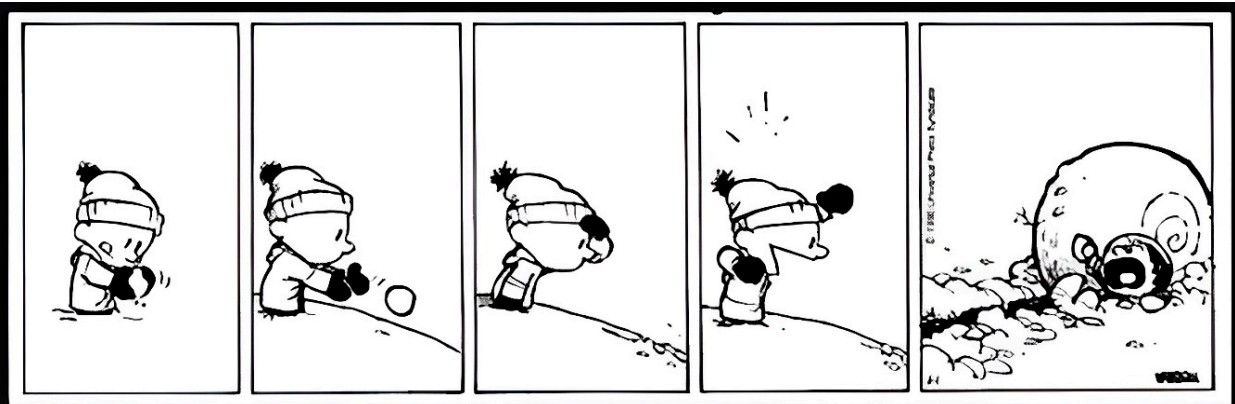

**Figure 3.** Visual metaphor of the "compound interest" concept. Source: www.medium.com. Available online: http://bit.ly/3HcxYvI (archived on 23 January 2023).

The third scene shows the person looking faraway. This is depicted by putting the hand on the forehead, which is the gesture for looking far by focusing at a point. This indicates the future time period that once the initial deposit has been submitted, then it takes some time to compound interest on it; it could be some or several months or years to reach an expected amount. The fourth scene shows joy and happiness, which is illustrated by the happy face, and the upward arm movement, which represents a winning body posture (Latu et al. 2017). This shows that the initial deposit has earned its whole term's compound interest and has reached a very high profit. The fifth and the last scene shows another person who has been stuck under a huge snowball. The facial emotions of this person showed pain due to the big snowball that ran over. This illustrates the negative aspect of compound interest, because there are usually two parties, one who earns the compound interest and the other who has to pay the interest amount. The last scene shows the one who has to pay for the compound interest. The negative aspect is that, if the person obtains a loan, the interest amount grows higher and higher to the point where it becomes a huge burden to pay back.

*3.4. Time Value of Money*

The time value of money (TVM) concept suggests that the value of an amount of money in-hand today is better than the same amount in the future. This is because of the earning potential of the sum of money that is in hand today that can be used to invest, on which returns can be gained in the interim. For example, USD 10,000 invested today at a 10% rate would yield USD 11,000 after the time period of one year. This situation is better than having USD 10,000 after one year, because in the meantime USD 10,000 could generate the return of an additional USD 1000. However, if the money is left without being invested, then it will not generate any potential returns. Thus, it is important to invest the money that is in-hand today. In fact, the buying power of the sum of money will reduce overtime due to the effect of inflation. The buying power is the quantity of goods and services that can be bought by the given sum of money. In this way, having USD 10,000 today and not investing the sum anywhere is equal to having the same amount after a year. It is only the earning potential based on the availability of time that makes it a rational decision to have money today and invest it.

Figure 4 illustrates several different elements as a metaphor of the time value of money concept. It shows a traditional weighing scale consisting of two bowls that are suspended at an equal distance from a fulcrum. One bowl has some money in it, while the other one has a time clock, which represents the time (Vuori 2010). These two bowls show that an individual has two options to choose from: either to have the money today, or to wait

and obtain the money later. The fulcrum is depicted by a hand demonstrating that the power of decision in in the individual's own hand. Technically, when either side of the weighing scale becomes heavier, it moves downwards. However, in this illustration, it shows that whichever direction the hand tilts towards, that side will move downwards, which represents the intentional selection of each decision. If an individual wishes to have money today, they can tilt their hand towards the right; if they wish to wait for a later period, then their hand may tilt to the other side. Nonetheless, the figure shows that the weighing scale is not tilted anywhere; it is, in fact, balanced. This balance expresses a rational thinking approach, in the same way that a balance is deemed to be rational in other aspects of life, such as work–life balance. It shows here that the individual opted for both the money and the time; depicting that there should not be a delay in receiving the money and investing it today to benefit from the available period of time.

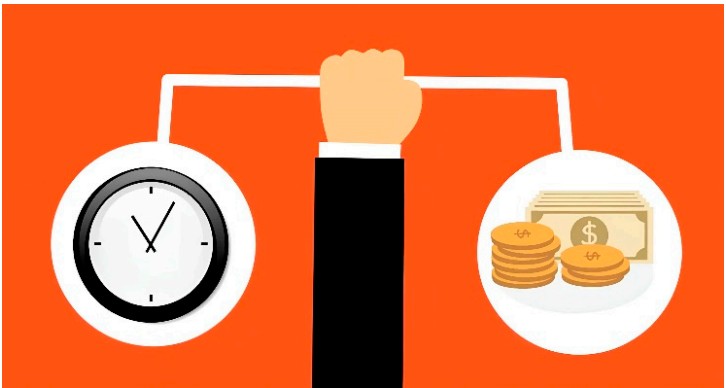

**Figure 4.** Visual metaphor of the "time value of money" concept. Source: www.magnimetrics.com Available online: https://bit.ly/3J2CQoj (archived on 8 March 2023).

*3.5. Financial Risk*

Financial risk is generally associated with the possibility of losing money for individuals, governments, or businesses. Most, if not all, financial investment decisions contain at least some amount of risk that poses a threat to the investment. There are various factors that are not always in control for the interested parties. The financial markets may be jeopardized due to macroeconomic elements, such as the 2007–2008 global financial crisis; organizations in which money is invested may default; or even pandemics, such as COVID-19, can completely block the economic activities. In the field of finance, financial risk is everywhere and it is present in different forms and shapes. There are several types of risks, some of the major types are: market risk, credit risk, liquidity risk, operational risk, and legal risk. Individuals usually face financial risk when they make financial decisions, which have the potential to endanger their returns or ability to repay the debt. It is crucial to consider the risks linked with financial decisions, and make sure to plan ahead if things go in the wrong direction; this would be then referred to as an informed or responsible financial decision. To have knowledge about the financial risks and learning the methods to protect oneself may not totally reduce the presence of such risks; however, it could, significantly reduce the chances of an unfavorable outcome of a financial decision.

Figure 5 represents the concept of a financial risk. It shows a man rope-crawling over water with a bag of money, under him; in addition, there are sharks in the water that are searching for their prey. The cash falling out of the bag shows that it is overfull of money. There are several mappings in this illustration related to financial risk, management of financial risk, and financial wellbeing. The figure shows that if one wishes to gain returns on an investment, then that person must cross this deadly water through a rope crawl. The sharks in the water refer to several types of financial risks posing threats. In order to dodge these sharks, one needs to have the skill of rope crawling. The rope crawl here demonstrates the ability to manage and avoid financial risk. In most cases, rope crawling needs training and practice to learn. In the same way, managing financial risk needs knowledge and

practical experience to become proficient. In the figure, there is a smile on the face of the man that shows he is capable of rope crawling (managing financial risk) to get past the sharks (financial risks). Furthermore, owing to the fact that most sharks are carnivorous, if a man does fall down, the sharks would be more interested in eating him instead of eating money in the bag. This represents that in situations when individuals do experience a financial loss, at times it may essentially harm their financial wellbeing, making them insecure of their financial future.

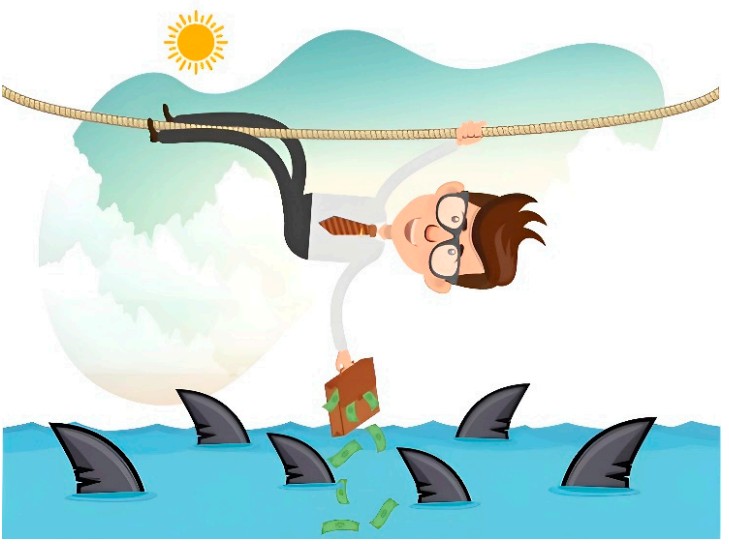

**Figure 5.** Visual metaphor of the "financial risk" concept. Source: www.capital.com Available online: https://bit.ly/3yETUvX (archived on 8 March 2023).

## 4. Applying Visual Metaphors to Learning

There could be several ways of applying the visual metaphors to the financial learning context; for this purpose, prior studies may provide some meaningful hints. In the study of Taylor et al. (2018), visual metaphors were merged into the digital storytelling, in which the accounting subject was used as a learning content. The aim was to assess the impact of digital storytelling on the comprehension and engagement of university students regarding accounting topics. Digital storytelling comprises a multimedia instructional design that aids learners to grasp complex topics in a captivating manner (Ivala et al. 2013). The storytelling content was created using the medium of video containing images, audio, animation, and text. The visual metaphors were inserted as animations throughout the storytelling video. The results revealed positive effects on students' learning, and in particular, the usage of visual metaphors amplified the impact of the digital storytelling. The authors suggested using the narrative and visual metaphors in lectures to incite engagement and comprehension.

Similarly, the study of Alyami et al. (2019) embedded visual metaphors into a serious game, which is an educational game that conveys knowledge in an entertaining way. The objective of the study was to analyze the impact of a serious game compared to a simple text file on learning about clinical history-taking of patients. For this purpose, the different components of the learning material in the game were presented through visual metaphors, which also contained text explanations. The study applied a quasi-experimental design, in which the treatment group was assigned to game learning, and the control group was assigned to the text file. Results showed that the participants in the treatment group found the visual metaphors based on a serious game more engaging and enjoyable.

While visual metaphors offer several benefits for learning, they may pose challenges as well. One of the major weaknesses is the potential for misunderstanding them (Savvas and Chilton 2019). More precisely, it could be explained as an error in recognizing the two domains being compared; this is owing to the implicit nature of metaphors. In

such situations, the desired learning objectives will most likely fail in transferring to the learners. This misunderstood interpretation may arise due to the lack of cultural competency, which is knowledge of the references used in the metaphor, or it could be attributed to the higher cognitive efforts required to understand the visual metaphor (McQuarrie and Mick 1999) that may overload the cognition. To minimize this challenge, the use of supplementary information to explain the visual metaphors has been found to be highly effective (Bergkvist et al. 2012).

## 5. Discussion and Conclusions

The findings are in the form of tentative conclusions inviting more research to study the representation of financial concepts through visual metaphors. The analysis in this study showed that there are varieties of ways through which financial concepts can be depicted using visual metaphors. For instance, the visual depiction of a verbal metaphor "don't put all your eggs in one basket" explains the complex financial concept of risk diversification in a simpler manner. This is because the verbal metaphor is commonly known and well understood, it is represented here in a visual depiction, where learners can pictorially relate its meaning with a complex concept. Compound interest is shown through a snowball effect. The snowball effect is a process that initially starts as being of little significance but it swiftly builds up and becomes larger and potentially disastrous as well. It is a cliché that is commonly shown in animated films or cartoons. Here, it represents the concept of compound interest. In all of these visual metaphors, there is one thing in common and that is the usage of concrete and/or commonly known phenomena as a source domain to associate it with a more complex and abstract financial concept that is a target domain. This is the fundamental idea of using metaphors to link the features and attributes of a known domain (source) onto an unfamiliar and complex domain (target) (Holyoak and Stamenković 2018).

The field of financial education could be improved using imaginative ways to generate relevant and representational pictures of financial concepts. The use of visuals can foster meaningful learning (Cañas and Novak 2014) that can, for instance, considerably increase retention and transfer (Schneider et al. 2016), but the usage of visuals in finance is largely confined to data graphs or charts only. The data visualization does not necessarily define the financial concepts, but merely shows the computations of them. For example, data visuals of inflation could be a graph of the last ten months' inflation rate. This graph may show how much inflation has been recorded in these months and what the trend was. However, these visuals will do little to essentially explain the concept of inflation itself. For these reasons, visual metaphors could be explored further in the financial context because they provide flexibility and variety to represent a complex concept along with precise linkages through the mappings of source and target domains (Ortiz 2011). Moreover, visual metaphors were found to increase engagement in the learning materials (Taylor et al. 2018), and they are effective for stimulating deeper understanding and communicating individual subjective experiences (Lazard et al. 2016).

One of the limitations of the study was the sample size, which was not representative of all the online visual metaphors used for improving financial literacy, because the sample was not big enough to represent the huge quantity of data of financial visual metaphors available online, nor was this aimed for. The metaphors in this paper were analyzed and discussed as case studies. The objective was to explore and gain a better understanding of the varieties and ways of how visual metaphors are developed for multifaceted financial concepts by educators and professional artists. Another limitation concerned the selected financial concepts that were related to investments and that, too, was not an exhaustive list. For future research, similar further analysis of representing financial concepts in the form of visual metaphors in various contexts is needed. It would also be useful to explore different methods of producing visual metaphors, in order to identify which method(s) are effective for producing these visuals of financial concepts. Lastly, experimenting with

visual metaphors to assess the impact on financial literacy in comparative contexts may generate valuable findings.

**Funding:** This research received no external funding.

**Data Availability Statement:** Publicly available data was analysed in this study. This data can be found on archived links provided in the manuscript.

**Conflicts of Interest:** The authors declare no conflict of interest.

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
