# Peer review of "Evaluating the Visual Metaphors of Financial Concepts through Content Analysis"

_jrfm, doi:10.3390/jrfm16030202_

Round 1

Reviewer 1 Report

The article examines the way in which images can be used to provide a visual metaphor of a complex and abstract concept within the finance context. The article proports to use content analysis to examine the way that images are being used. Specifically, it examines one image related to five of the most critical and basic financial literacy concepts examining how the image can be related to the underlying concept.

I applaud the authors intention to examine ways in which financial education can be more effectively delivered. To date, the evidence on the success of financial education as a tool to improve financial literacy and by extension financial decision making is mixed but leans towards questioning it's efficacy. We critically need to understand why efforts to improve financial education are less successful than we might hope, and what we can do to improve it. The use of visual metaphors offers a potential avenue.

While I applaud the author(s) for venturing into this area, I expected the paper to do slightly more than it did, and in its current form it makes a limited contribution. This paper simply seems to examine the question of whether abstract complex financial concepts can be expressed in a visual form. While this establishes whether it can be done, it is also something of a redundant question given that finance academics have been using similar images or concepts to illustrate these concepts for some time, especially in relation to the use of snowballs for compound interest and baskets of eggs for diversification. The article left me with the question of whether these images are effective in improving learning, or what the most effective images might be.

I also have several additional comments for the authors:

What was the value in using the ALL tab to search for educational webpages that contain images rather than doing an image search for the same search term? It would be good to see some justification for this.

In several cases I am not sure I agree with the degree to which the image is interpreted. There seems to be an effort to explain every aspect of the image. In the case of a custom designed image this might be valid, in the case of using repurposed images there is a risk of over-interpreting the scene. A case in point is the interpretation of the snowball metaphor to explain compounding. Compounding, when explaining the concept to lay people usually focuses on explaining the positive side of compounding. In this case, a snowball makes perfect sense. The image selected has five frames, the final frame shows a person run over by the snowball at the bottom of the hill. The author(s) have assumed this to mean that the image is showing the negative consequence of compounding as it applies to debt. The image to me looks like a repurposed cartoon in which case it is potentially doubtful that the webpage had that intent, rather it may have been incidental, or it may have been used to give scale to the effect that compounding can have.

This is similar to the TVM image which shows a small sum of money further away and a larger sum closer up. There is a line linking the two. The author(s) attempt to interpret that line bi-directionally, arguing that it shows how investing a small amount in the past can grow to a large amount in the present, while if reversed it shows that a large amount can be devalued over time. For a start the later interpretation is more a reference to inflation than TVM. Second, when I view the same image I don't see that at all. I see the TVM growing argument given the smaller sum is in the background and the larger sum in the foreground. This suggests a weakness in the authors method. It appears that there has been, for want of a better term, a single coder. What might be more effective is to have several coders and focus on the elements that both select as more indicative of the intent. It would be good to see some acknowledgement of these limitations of the study, specifically, that it is a single persons interpretation (if that is the case) and that there is a risk, especially if the authors are finance experts, of interpreting the image in the way that a lay person might not. Misconstruing the image would suggest that the image is not an effective way of conveying complex concepts without some additional material to provide a shared understanding.

Finally, the explanation of risk is not one I agree with. It focuses on the negative side of risk which in some respect correlates more with a common understanding of the term risk. In finance risk is a more ambivalent concept, it is neither a positive nor a negative. It is something that should be recognised and at times managed, but without risk there is no reward. I would like to see a more nuanced discussion of the concept of risk.

Author Response

Dear Reviewers and Editors,

As per your suggestions, I have revised the manuscript, titled: “Evaluating the Visual Metaphor of Financial Concepts Through Content Analysis”. This letter briefly mentions the major changes made to the paper. The revision has been done using the ‘Track Changes’ option in the MS Word, so that it would be convenient for the reviewers or editors to follow the new changes.

Concerning the revision, several changes have been made in different sections of the paper. Starting from the introduction section, more information has been added regarding the need of conducting this study. The aim of the paper is also stated clearly at different key places, like introduction, methodology, and conclusion. In the methodology section, it has been discussed in detail regarding the background, scope, and limitations of the analysis. Also, this section reflects on how these analyses are in line with the objective of the paper, and what implications does it provide. Other than the analysis, further information has been added to the selection of the sample. In the section of interpreting visual metaphors, for clearer understanding, two visual metaphors have been replaced, which include the concepts of ‘time value of money’ and ‘financial risk’. As per one of the reviewers, these two were not so clear. After the analysis section, a new section by the title of ‘applying visual metaphors to learning’ has been added. This was recommended by one of the reviewers by giving reasons that, to show a weakness of visual metaphors and the ways regarding how they can be applied to the learning context. Finally, there have been some changes made to the implications of the study. These implications have been discussed under the explicit mention of the objective, scope, and limitations of the study. 

Reviewer 2 Report

I think this is a nice paper with an original idea and it could be published perhaps after minor revisions. Addressing visual representations on financial literacy is a good idea. The topics are well selected. I especially appreciated the snowballing / compound interest metaphor and the associated discussions. The paper is well written and clearly demonstrates familiarity with the topics. Some issues to be considered:

1) I think the limits of the metaphors could be explained. In some cases, the metaphor is to the point but the potential for learning may be limited (like the time value example). In other cases, the metaphor may not capture all essential elements. For instance, in the "all eggs in the same basket" / diversification example it would appear that diversification might be somewhat high in transaction costs (you need separate baskets for eggs). In reality, diversification may entail even a free lunch (higher expected value with lower risk), not really captured by the visual metaphor.

2) You could also extend the discussion from single visual images to cartoons (although some of these are short strips). By cartoons, you could actually provide richer lessons and have a richer media, while at the same time retaining the visual metaphor.

3) The visual metaphors may not be self-evident. How are the learners expected to use the pictures? Probably they need a teacher or expert guidance, like here, where you have an expert commentary on the metaphors by the author. I would appreciate some discussion on applying the visual metaphors in actual learning situations.

4) On p. 5 you identify money supply as the root case of the inflation. This may not be the only cause, as supply and demand of other commodities than money may also be affecting inflation. However, your explanation is consistent with the picture, where there seems to be only one factor influencing inflation.

Author Response

(The authors gave the same response as above.)
